# Drug-Induced Naïve iPS Cells Exhibit Better Performance than Primed iPS Cells with Respect to the Ability to Differentiate into Pancreatic β-Cell Lineage

**DOI:** 10.3390/jcm9092838

**Published:** 2020-09-02

**Authors:** Yuki Kiyokawa, Masahiro Sato, Hirofumi Noguchi, Emi Inada, Yoko Iwase, Naoko Kubota, Tadashi Sawami, Miho Terunuma, Takeyasu Maeda, Haruaki Hayasaki, Issei Saitoh

**Affiliations:** 1Division of Pediatric Dentistry, Faculty of Dentistry & Graduate School of Medical and Dental Sciences, Niigata University, Niigata 951-8514, Japan; ykiyokawa@dent.niigata-u.ac.jp (Y.K.); iwase@dent.niigata-u.ac.jp (Y.I.); hayasaki@dent.niigata-u.ac.jp (H.H.); 2Section of Gene Expression Regulation, Frontier Science Research Center, Kagoshima University, Kagoshima 890-8544, Japan; masasato@m.kufm.kagoshima-u.ac.jp; 3Department of Regenerative Medicine, Graduate School of Medicine, University of the Ryukyus, Okinawa 903-0215, Japan; noguchih@med.u-ryukyu.ac.jp; 4Department of Pediatric Dentistry, Kagoshima University Graduate School of Medical and Dental Sciences, Kagoshima 890-8544, Japan; inada@dent.kagoshima-u.ac.jp (E.I.); kubonao@dent.kagoshima-u.ac.jp (N.K.); 5Yokohama City Center for Oral Health of Persons with Disabilities, Kanagawa 231-0012, Japan; sawami1972.dent@gmail.com; 6Department of Oral Biochemistry, Faculty of Dentistry & Graduate School of Medical and Dental Sciences, Niigata University, Niigata 951-8514, Japan; mterunuma@dent.niigata-u.ac.jp; 7Center for Advanced Oral Science, Faculty of Dentistry & Graduate School of Medical and Dental Sciences, Niigata University, Niigata 951-8514, Japan; maedat@dent.niigata-u.ac.jp

**Keywords:** induced pluripotent stem cells, pancreatic β-cells, pancreas, naïve stem cells, drug induction, differentiation, epiblast stem cells, insulin-producing cells, intrapancreatic parenchymal cell transplantation (IPPCT), teratoma formation assay

## Abstract

Pluripotent stem cells are classified as naïve and primed cells, based on their in vitro growth characteristics and potential to differentiate into various types of cells. Human-induced pluripotent stem cells (iPSCs, also known as epiblast stem cells [EpiSCs]) have limited capacity to differentiate and are slightly more differentiated than naïve stem cells (NSCs). Although there are several in vitro protocols that allow iPSCs to differentiate into pancreatic lineage, data concerning generation of β-cells from these iPSCs are limited. Based on the pluripotentiality of NSCs, it was hypothesized that NSCs can differentiate into pancreatic β-cells when placed under an appropriate differentiation induction condition. We examined whether NSCs can be efficiently induced to form potentially pancreatic β cells after being subjected to an in vitro protocol. Several colonies resembling in vitro-produced β-cell foci, with β-cell-specific marker expression, were observed when NSC-derived embryoid bodies (EBs) were induced to differentiate into β-cell lineage. Conversely, EpiSC-derived EBs failed to form such foci in vitro. Intrapancreatic grafting of the in vitro-formed β-cell foci into nude mice (BALB/c-nu/nu) generated a cell mass containing insulin-producing cells (IPCs), without noticeable tumorigenesis. These NSCs can be used as a promising resource for curing type 1 diabetes.

## 1. Introduction

Type 1 diabetes is caused by autoimmunity against insulin-producing β-cells in pancreatic islets of Langerhans. The incidence of Type 1 diabetes shows wide variation across different countries [1]. Shapiro et al. developed a technique for islet transplantation with glucocorticoid-free immunosuppression [2]. However, Ryan et al. demonstrated that among the patients who had islet transplantation, only 10% of patients could maintain exogenous insulin independence 5 years after the islet transplantation [3]. Many patients with Type 1 diabetes control their blood glucose levels by injecting insulin several times daily because of the poor therapeutic options currently available. 

In 2007, it was reported that among the mouse embryonic stem cells (ESCs), there are at least two types of ESCs, namely the ground or naïve state cells [also known as naïve stem cells (NSCs)] and primed pluripotent state cells, which are now called epiblast stem cells (EpiSCs)] [4,5]. The differences between the two cell types include colony morphology, growth factor requirement for maintenance of the pluripotent state, and X inactivation status in female cells [6,7]. Human ESCs are thought to be more closely related to mouse EpiSCs [8]. Human-induced pluripotent stem cells (iPSCs, also known as EpiSCs) are generated by reprogramming adult somatic cells with four reprogramming factors (also known as OSKM Yamanaka factors [9]), namely octamer-binding transcription factor-3/4 (*OCT-3/4*), sex-determining region Y-box 2 (*SOX2*), Krüppel-like factor-4 (*KLF4*), and c-myelocytomatosis (*c-MYC*) [8]. The EpiSCs have unlimited potential to self-renew and differentiate into three germ layers in vitro but have limited pluripotency ability in vivo [10].

In 2008, Ying et al. demonstrated that serum-free synthetic medium containing 2i [PD0325901 (an inhibitor of the MEK/ERK pathway), CHIR99021 (an activator of the Wnt/β-catenin signaling pathway by inhibiting the activity of glycogen synthase kinase 3)], and leukemia inhibitory factor (LIF) is beneficial in maintaining mouse ESCs in a naïve state (or ground state) [11]. Hanna et al. reported that NSCs could be successfully induced when human iPSCs were treated with reprogramming-related drugs, such as 2i, kenpaullone (an inhibitor of glycogen synthase kinase 3), or forskolin (known to increase intracellular levels of cyclic AMP) [12]. According to these researchers, the resulting NSC-like cells resembled the property of murine ESCs, as demonstrated by their dome-like colony morphology, rapid proliferation, higher survival rate after trypsinization (which indicated lower dependency on the Rho-associated coiled-coil forming kinase inhibitor), and activation of both X-chromosomes in female NSCs. Since then, several methods to obtain NSCs have been reported. Kilens et al. transfected somatic cells with Yamanaka factors, and subsequently cultivated these cells under specific culture conditions [13]. Takashima et al. treated EpiSCs with a cocktail-containing medium called t2iLGö, which consisted of CHIR99021, PD0325901, basic fibroblast growth factor (bFGF) (also known as FGF2), and Gö6983 [14]. Theunissen et al. used a cocktail called 5i/L/AF which contained 5i (PD0325901, CHIR99021, SB590885, WH-4-023, and Y-27632), supplemented with LIF, bFGF, and activin A [15]. These methods suggested that chemical induction was more convenient to obtain NSCs, owing to its simplicity and the use of pre-existing EpiSCs.

Using the method described by Hanna et al. [12], we recently demonstrated that human deciduous teeth dental pulp cells (HDDPCs)-derived iPSCs (which are hereinafter referred to as HDDPC-EpiSCs) could be successfully converted to NSC-like cells (which are hereinafter referred to as HDDPC-NSCs), which possessed specific properties such as accelerated growth rate, dome-like colony morphology, expression of ESC-associated stem cell-specific markers [as exemplified by zinc finger protein 42 (ZFP-42) or REX-1], expression of stage-specific embryonic antigen 1 (SSEA-1), and the ability to differentiate into mouse vasa homolog-positive cells after teratoma formation in vivo [10]. When a similar approach was adopted using HDDPC-EpiSCs, we found that HDDPC-EpiSCs had poorer differentiation potency than NSCs, as there were relatively fewer types of differentiated cells in the former [10]. This suggested us to evaluate whether both EpiSCs and NSCs would behave differently when induced to differentiate into pancreatic β-cell lineage. Here, we examined this possibility using drug-induced HDDPC-NSCs and their parental cells HDDPC-EpiSCs. Our results demonstrated that HDDPC-NSCs efficiently generated intermediate cells to differentiate into pancreatic β-cells in vitro and expressed pancreatic-duodenal homeobox factor-1 (*PDX1*) and insulin.

## 2. Experimental Section

### 2.1. Ethical Approval

Induction and cultivation of HDDPC-EpiSCs, HDDPC-NSCs, and their differentiated derivatives: induced tissue-specific stem cells for pancreatic (iTS-P) cells, were performed according to the guideline and the protocol approved by the Ethical Committee for the Use and Experimentation of Graduate School of Medical and Dental Science, Niigata University (No. 2017-0185; dated on 16 October 2017). In addition, the experiments described in this study were performed in agreement with the guidelines of Niigata University Committee on Recombinant DNA Security (No. SD00798; dated on 4 October 2017) and with the approval by the Animal Care and Experimentation Committee of Niigata University (No. SA00150; dated on 7 December 2017).

### 2.2. Induction of HDDPC-NSCs

The HDDPC-EpiSCs, generated using our own protocol, were used for this study as a parental cell line for obtaining HDDPC-NSCs. HDDPC-EpiSCs were grown on the mitomycin C (MMC)-treated (#M4287; Sigma-Aldrich, St Louis, MO, USA) mouse embryonic fibroblast (MEF) cells in a 60-mm gelatin-coated dish (#4010-020; Iwaki Glass, Tokyo, Japan) with human embryonic stem cell culture medium, iPSellon (#007001; Cardio, Kobe, Japan), supplemented with 5 ng/mL recombinant human bFGF (#064-04541; Wako Pure Chemical Industries, Ltd., Osaka, Japan). Cell passage was performed on the fifth day after cell seeding on MEF cells. Half of the medium was replaced with fresh medium every day. 

For induction of HDDPC-NSCs, HDDPC-EpiSCs were cultured on the MMC-treated MEF cells in a 60-mm gelatin-coated dish with NSC medium, based on N2B27 medium (also called NDiff^®^ 227; #Y40002; Takara Bio Inc., Shiga, Japan), containing 5 ng/mL of recombinant human bFGF, 0.02 µg/mL of recombinant human LIF (#L5283; Sigma-Aldrich), 1 µM of PD0325901 (#162-25291; Wako Pure Chemical Industries, Ltd.), 10 µM of PD98059 (#169-19211; Wako Pure Chemical Industries, Ltd.), 3 µM of CHIR99021 (#038-23101; Wako Pure Chemical Industries, Ltd.), 10 µM of forskolin (#067-02191; Wako Pure Chemical Industries, Ltd.), and 1 µM of kenpaullone (#110-00831; Wako Pure Chemical Industries, Ltd.). Medium change and cell passage were done using the similar procedures as described for the cultivation of HDDPC-EpiSCs.

### 2.3. Reverse Transcription-Polymerase Chain Reaction and Densitometric Image Analysis 

Total RNA from NSCs and iTS-P was isolated using an RNA mini kit (#50204; QIAGEN N.V., Venlo, Limburg, The Netherlands). To identify the expression of target mRNA by reverse transcription-polymerase chain reaction (RT-PCR), reverse transcription was first performed using a first-strand cDNA synthesis kit (#18080-051; Invitrogen Co., Carlsbad, CA, USA). The resultant cDNAs were then PCR-amplified from undiluted cDNA samples (2 µL) in a total volume of 25 µL, using AmpliTaq Gold^®^ 360 Master Mix (#4398881; Applied Biosystems, Foster City, CA, USA). Primer sets are shown in Appendix A.

PCR was performed for 38 cycles of 30 s denaturation at 95 °C, 30 s annealing at 58 °C, and 60 s extension at 72 °C in a SimpliAmp Thermal Cycler (Applied Biosystems). The PCR products (5 µL each) were analyzed by 2% agarose gel electrophoresis, and subsequently stained with SYBR Safe DNA Gel Stain (#S33102; Invitrogen Co.). Images of stem cells and pancreatic markers were saved as TIFF files for semi-quantitative analysis as shown below. 

Each band in the PCR images was scanned and analyzed using ImageJ software (National Institutes of Health; http://rsbweb.nih.gov/ij/). The densitometric data of each transcript were normalized with those of the internal control, the *GAPDH* mRNA, and the results are expressed in graphs, according to Chapman et al. [16]

### 2.4. Induced Differentiation into Pancreatic β-Cell Lineage 

Differentiation into insulin-producing cells was performed as previously described [17,18], with minor modifications.

For embryoid bodies (EB) formation, cell colonies (>300), generated 5 days after seeding, were mechanically separated from the surface of a tissue-culture dish by removing the medium using a pipette tip or by removing the cells with a cell scraper (#3010; Corning Inc., New York, NY, USA), and left for 2 days to allow the formation of tightly packed cell aggregates. In this case, no medium change was done. Then, cell aggregates were collected by centrifugation at 1000 rpm for 5 min and the resultant cell pellet was suspended in Dulbecco’s modified Eagle medium (DMEM) (#11995-081; Invitrogen Co.)- fetal bovine serum (FBS) (#SFMB30-2239; Equitech Bio Inc., Kerrville, TX, USA) (DMEM-FBS), prior to cultivation on an ultralow attachment 35-mm dish (#MS-9035X; Sumitomo Bakelite Co., Ltd., (Tokyo, Japan) for 5 days at 37 °C in an atmosphere of 5% CO_2_ in air. After cultivation, the resultant EBs were seeded onto a 35-mm tissue-culture dish (#4000-020; Iwaki Glass Co., Tokyo, Japan) to them to promote outgrowth in DMEM-FBS for 2 days. Next, these cells were subjected to a stepwise protocol [17,18] to drive differentiation toward IPCs, as shown below and in Appendix A. 

In Stage 1, the cells were treated with 25 ng/mL Wnt3a (#1324-WN-002; R&D Systems, Inc., Minneapolis, MN, USA) and 100 ng/mL activin A (#338-AC-050; R&D Systems, Inc.) in RPMI medium (#30-2001; ATCC, Manassas, VA, USA) for 1 day, followed by treatment with 100 ng/mL of activin A in RPMI + 0.2% FBS for 2 days. 

In Stage 2, the cells were treated with 50 ng/mL fibroblast growth factor 10 (FGF10) (#6224-FG-025; R&D Systems, Inc.) and 0.25 μM 3-Keto-N-(aminoethyl-N′-aminocaproyldihydrocinnamoyl) cyclopamine (KAAD-cyclopamine) (#K171000; Toronto Research Chemicals, North York, ON, Canada) in RPMI + 2% FBS for 3 days. 

In Stage 3, the cells were treated with 50 ng/mL FGF10, 0.25 μM KAAD-cyclopamine, and 2 μM all-*trans* retinoic acid (#R2625; Sigma-Aldrich) in DMEM + 1% (vol/vol) B27 supplement (#0050129SA; Invitrogen Co.) for 3 days. 

In Stage 4, the cells were treated with 1 μM N-[N-(3,5-difluorophenacetyl)-L-alanyl]-S-phenylglycine t-butyl ester (DAPT) (#D5942; Sigma-Aldrich) and 50 ng/mL exendin-4 (#E7144; Sigma-Aldrich) in DMEM + 1% (vol/vol) B27 supplement for 3 days. 

In Stage 5, the cells were treated with 50 ng/mL exendin-4, 50 ng/mL insulin-like growth factor 1 (IGF-1) (#I1146; Sigma-Aldrich), and 50 ng/mL hepatocyte growth factor (#315–23; PeproTech Inc., Rocky Hill, NJ, USA) in Connaught Medical Research Laboratories medium (#11530–037; Invitrogen Co.) + 1% (vol/vol) B27 supplement for 3–6 days. The resultant iTS-P cells were continuously maintained in NSC medium on feeder layers of MMC-treated MEF cells.

### 2.5. Teratoma Formation/Tumorigenicity Assay

To induce solid tumor formation in vivo, NSC-like colonies (~300) or NSCs-derived intermediate cells (~300) were harvested by simple pipetting or trypsinization, and dissolved in 20 μL of iPSellon culture medium containing 2 μL of 0.4% trypan blue (#15250-061; Invitrogen Co.). Approximately 2 μL of the solution was then injected into the pancreatic parenchyma of nude female mice (BALB/cAJcl-nu/nu; 10–15 weeks old; CLEA Japan Ltd., Tokyo, Japan) using a glass micropipette (connected to the mouthpiece), under a dissecting microscope, according to Sato et al. [19] and Inada et al. [10]. The emerging teratomas (~1.5 months after grafting) or small lumps (6 months after grafting) generated were dissected and fixed with 4% paraformaldehyde (PFA) in Dulbecco’s modified phosphate-buffered saline, without Ca^2+^ and Mg^2+^ [D- phosphate-buffered saline (PBS)(-)], at 4 °C for one week. The fixed tissues were then dehydrated by immersion in 0.25% sucrose in D-PBS(-) at 4 °C for two days, and then dehydrated in 0.4% sucrose in D-PBS(-) at 4 °C for four days. These samples were then embedded in optimum cutting temperature compound Tissue-Tek^®^ (#4583; Miles Scientific, Naperville, IL, USA) for cryostat sectioning (5 µm in thickness). Some cryostat sections were stained with hematoxylin and eosin, while others were subjected to immunostaining with antibodies, as described below.

### 2.6. Immunostaining and Cytochemical Staining

Cells were fixed with 4% PFA in D-PBS(-) at 4 °C for 10 min. After blocking with 20% AquaBlock (#PP82; East Coast Bio, North Berwick, ME, USA) for 30 min at 24 °C, the cells were incubated for 12 h at 4 °C, with a goat anti-insulin primary antibody (1:100; #ab7842; Abcam, Tokyo, Japan) or goat anti-*PDX1* primary antibody (1:100; #ab47383; Abcam) and then for 1 h at 24 °C with FITC-conjugated goat IgG (1:250; #ab6904; Abcam) or Alexa Fluor 488-conjugated anti-rabbit IgG (1:250; #ab150077; Abcam). For cytochemical staining for ALP activity, ALP Phosphatase Staining Kit II (#00–0055; STEMGENT, Cambridge, MA, USA) was used, according to the protocol indicated by the manufacturer. 

Cryostat sections were first incubated in 20% AquaBlock for 30 min at 24 °C, and then incubated for 12 h at 4 °C with the above-mentioned antibodies as well with anti-C-peptide primary antibody (1:100; #4593; Cell Signaling). After washing with D-PBS(-), the same specimens were incubated for 12 h at 4 °C with human cells-specific mouse monoclonal antibody STEM121 (1:300; #Y40410; Cellartis–Takara Bio, Kusatsu, Japan), followed by incubation in 20% AquaBlock for 30 min at 24 °C. After washing, the specimens were incubated for 1 h at 24 °C with Alexa Fluor 647-conjugated anti-rabbit IgG (1:250). 

The cells and sections were treated with mounting medium for fluorescence with 4′,6-diamidino-2-phenylindole (DAPI) (#H-1200; Vector Laboratories, Burlingame, CA, USA). 

### 2.7. Fluorescence Observation 

Fluorescence was examined using a fluorescence microscope (Olympus BX60; Olympus, Tokyo, Japan). Microphotographs were obtained using a digital camera (FUJIX HC-300/OL; Fuji Film, Tokyo, Japan), attached to the fluorescence microscope, and printed using a Mitsubishi digital color printer (CP700DSA; Mitsubishi, Tokyo, Japan). 

### 2.8. Statistical Analysis

Data are expressed as the mean ± SE. The Student *t* test was used to compare the densitometric data of each transcript between NSC and iTS-P. *p* < 0.05 indicates significant difference.

## 3. Results

Figure 1 shows the schematic diagram of the overall experimental procedures followed in this study. To stimulate in vitro differentiation of NSCs and EpiSCs into pancreatic β-cell lineage, HDDPC-EpiSCs and HDDPC-NSCs were first subjected to EB formation. This appears to be the first trigger to stimulate endodermal differentiation. Next, these EBs were subjected to a stepwise differentiation induction toward pancreatic β-cell lineage. During these steps, the cells were inspected for their morphology and analyzed by in vivo teratoma formation assay (for HDDPC-NSCs alone). At the final stage of differentiation, the cells were subjected to morphological inspection, immunocytochemistry, and RT-PCR analysis for the detection of β-cell-specific markers and analyzed by the in vivo teratoma formation assay (for HDDPC-NSCs alone).

### 3.1. Formation of Drug-Induced HDDPC-NSCs

HDDPC-NSCs were obtained by culturing HDDPC-EpiSCs [20] in NSC medium, containing reprogramming-related drugs [2i (PD0325901 + CHIR99021), kenpaullone, and forskolin] in a 60-mm dish with MEF-derived feeder cells, at 37 °C in an atmosphere of 5% CO_2_/95% air (Figure 2A). After passage 10, the NSCs could be identified as cell aggregates that were dome shaped (Figure 2A(a,b)), expressed alkaline phosphatase (ALP; one of the stemness markers) (Figure 2A(c)), and presented increased self-renewal capacity, as described previously [10]. Concomitantly, EpiSCs were cultured in the human ES cell culture medium iPSellon containing recombinant human bFGF. These cells were flat (Figure 2A(d,e)) and expressed ALP (Figure 2A(f)), as described previously, and they were used as the control in this experiment.

### 3.2. Differentiation Induction toward β-Cell Lineage

For EB formation, cell colonies (>300), generated 5 days after seeding, were mechanically separated from the surface of a tissue-culture dish by removing the medium using a pipette tip or by removing cells with a cell scraper, and left for 2 days to allow the formation of tightly packed cell aggregates. In this case, no medium change was performed. Subsequently, the cell aggregates were collected after a brief centrifugation and the resultant cell pellet was suspended in DMEM, supplemented with 10% FBS (which is hereinafter referred to as DMEM-FBS). An aliquot of the solution, containing approximately 100 aggregates was transferred to a non-coated dish and cultured for 5 days (Figure 1). After cultivation of the cell suspension, the resulting EBs (Figure 2B(a,d); ~50) were seeded onto a 35-mm tissue-culture dish (containing DMEM-FBS) to allow them to promote their outgrowth onto the substratum for 2 days (Figure 1). Subsequently, these cells were subjected to a stepwise protocol [17,18] to drive differentiation toward insulin-producing cells (IPCs) (Figure 1; Appendix A). 

First, NSCs (or EpiSCs) were allowed to differentiate into definitive endoderm (DE) by cultivation in a medium containing activin A for 3 days, defined as Stage 1. In this case, Wnt3a was added on the first day of incubation. 

In Stage 2, to induce transition from definitive endoderm to primitive gut tube (PG), the medium was changed to a medium containing FGF10 and hedgehog-signaling inhibitor KAAD-cyclopamine, and cultivated for 3 days.

In Stage 3, to induce transition from PG to posterior foregut (PF), the cells were cultivated in a medium containing retinoic acid, KAAD-cyclopamine, and FGF10 for 3 days. During this time, intermediate stem cells (also known as induced tissue-specific cells or induced tissue-specific pancreatic stem cells [21]) were thought to appear. These cells were hereinafter called as “induced tissue-specific stem cells for pancreatic cells (iTS-P)” [22], since they were assigned as pancreatic precursor cells. According to Saitoh et al. [22], these cells formed aggregates with each other and showed “cobblestone”-like colonies with smooth surface. Indeed, in the group in which the NSCs were cultivated, several small colonies with flat surface were discernible (arrows in Figure 2B(b)). In contrast, no noticeable colony was observed when EpiSCs were cultivated up to Stage 3 (Figure 2B(e)).

To advance the differentiation toward IPCs, the cells in Stage 3 were further cultured in a medium containing DAPT (a γ-secretase inhibitor), exendin 4, IGF-1, and bFGF, along with 27 supplements, for 6 days to induce pancreatic endoderm and endocrine precursors, defined as Stages 4 and 5. As a result, in the group in which the NSCs were induced to differentiate, there were several colonies showing “cobblestone” morphology (arrows in Figure 2B(c); also shown as enlarged figure in the quadrant of the right bottom of Figure 2B(c)) at Stage 5. Notably, the morphology of the “cobblestone”-like colonies resembled that of mouse iTS-P [22]. Thus, we hereafter called these cobblestone-like colonies “iTS-P.” An average of 20 iTS-P per dish (among the three dishes examined) were observed. Particularly, iTS-P could be maintained in NSC medium beyond 30 passages. In contrast, there were no noticeable colonies in the group in which the EpiSCs were treated for differentiation into pancreatic β-cell lineage (Figure 2B(f)). 

### 3.3. Pancreatic Marker Gene Expression in NSCs-Derived Cells

We investigated the mRNA expression of stem cell-specific or pancreatic marker genes in NSCs and NSC-derived iTS-P (collected at 16 days after the induction of differentiation into pancreatic β-cells), using the RT-PCR method (Figure 3A). Both NSCs and iTS-P expressed stem cell-specific markers, such as *OCT-3/4* and *SOX2*. In addition, iTS-P expressed pancreatic markers, such as *PDX1* and insulin. In Figure 3B, the intensity of each band shown in Figure 3A was normalized with that of *GAPDH* mRNA, and the results are expressed in graphs. Clearly, iTS-P cells exhibited 2.4- and 8.5-fold significantly higher expression of insulin and PDX1, respectively, than their parental NSCs, in the densitometric image analysis.

Immunocytochemical staining further confirmed the results obtained in RT-PCR analysis. iTS-P cells at Stage 5 that differentiated from NSCs and formed cobblestone-like colonies exhibited reactivity to both anti-*PDX1* and anti-insulin antibodies (“NSC-derived cells at Stage 5” in Figure 3C). In contrast, in EpiSC-derived cells at Stage 5 (Figure 3C), there were some single cells showing reactivity to both antibodies, but cell aggregates, similar to the cobblestone-like colonies, were absent.

### 3.4. Intrapancreatic Grafting of NSCs-Derived Cells Leads to the Generation of Insulin-Positive Cell Mass 

Determination of whether NSCs-derived iTS-P at Stages 3 and 5 were tumorigenic and had the ability to form islet like 3-D structure was still a challenge. To resolve this issue, in vivo grafting of iTS-P into appropriate tissues or organs of immunocompromised mice is an ideal strategy, as suggested by Nelakanti et al. [23]. In this context, it would be better to use our recently developed novel method (called “intrapancreatic parenchymal cell transplantation (IPPCT)”), because it enables the growth of a small number of tumor cells (including EpiSCs and NSCs) through injection of cells into pancreatic parenchyma of a nude mouse (BALB/c-nu/nu) [10,19]. Using this method, we have succeeded in generating solid tumors (teratomas), measuring approximately 1 cm in diameter, within 1.5 months after grafting of EpiSCs [19]. 

As shown in Figure 4A, NSC-derived iTS-P (~60) at Stages 3 and 5 (after passage 15) or NSCs (~10^3^) (used as control) were subjected to IPPCT. As for EpiSC-derived cells, we failed to generate obvious colonies after in vitro induction of differentiation, as shown in Figure 2B(f). Furthermore, in our previous study, we found that the formation of cobblestone-like colony is a prerequisite for forming cellular mass capable of secreting insulin in vivo. This indicated the inability to generate iTS-P and their derivatives with a highly organized structure when EpiSCs were used as the starting material for isolating pancreatic β cells. Because of this reason, we did not continue with the grafting of these in vitro cultured EpiSCs derivatives. After grafting, the mice were carefully examined once a week to observe for any sign of tumorigenesis, which could be easily detected by the expansion of ventral portion above the pancreas. In case of grafting of NSCs, tumor formation was evident 1.5 months after grafting. Thus, we decided to dissect the generated solid tumor from the pancreas and immediately subjected it to fixation and subsequent cryostat sectioning. Histological examination revealed that there were many types of differentiated cells, such as glandular structure, digestive duct-like cells, and cartilaginous tissues (Figure 4B(a–e)). In contrast, in the case of grafting of NSCs-derived iTS-P cells at Stages 3 and 5, no significant swelling in the ventral skin was noted even after 1.5 months post-surgery. Therefore, the time for tissue sampling was extended by 6 months after grafting. Upon dissection at 6 months after grafting, small masses (approximately <5 mm in diameter), that did not look like solid tumor, were visible as a whitish lump in the pancreas of both the grafts (right panels in Figure 4A).

Cellular masses containing organized structures were observed in both the grafts. However, the graft from NSC-derived iTS-P, isolated at Stage 5, had more highly organized structure, with glandular epithelial cells, compared to that in iTS-P isolated at Stage 3 (Figure 4B(f,g) vs. Figure 4B(h–j)). Immunostaining of these cryostat sections (Figure 5A(a–h) and Figure 5B(a–l)) using anti-insulin antibody, demonstrated that both the specimens (derived from grafting iTS-P at Stages 3 and 5) were reactive to the antibody (Figure 5A(b),B(b)). Notably, in the sample derived from grafting of iTS-P at Stage 5, anti-insulin staining of a highly organized structure, containing glandular cells (Figure 5B(e,f)) and islet-like structure (Figure 5B(i,j)), was observed. We suspected that the latter cells might be the remnant of host (mouse)-derived pancreatic β-cells, because the anti-insulin antibody used is known to react with both human and mouse IPCs (as per the manufacturer’s instruction). To distinguish the cells derived from the human graft from those derived from the endogenous (host) pancreatic β-cells, double labeling was performed. We stained cryostat sections with anti-insulin antibody, again with human cells-specific mouse monoclonal antibody STEM121. In glandular cells with extensive staining by anti-insulin antibody (arrowhead in Figure 5B(j–l)), the intracellular portion of these cells was also stained with STEM121 (arrows in Figure 5B(j–l)), suggesting that these insulin-positive cells were indeed of human origin. In the islet-like structure, which was located near the glandular cells (as mentioned previously), the outer surface of the islets was stained with STEM121 (arrowhead in Figure 5B(f–h)). However, the intracellular portion of these cells was not stained with STEM121 (arrows in Figure 5B(f–h)), suggesting that these islet-like structures were of mouse (host) origin. Furthermore, since the outer surface of the islets was stained with STEM121, this may suggest that human-derived cells had already surrounded these islet-like structures. Immunostaining using anti-C-peptide antibody demonstrated that both the specimens, derived from grafted iTS-P at Stages 3 and 5, were reactive to the antibody (Figure 6). These findings suggest the possible production of insulin using these tissues. Furthermore, both the specimens were positively stained by anti-*PDX1* antibody (Appendix A).

We also immunohistochemically examined whether solid tumors generated from grafting of NSCs have IPCs. Cryostat sections (shown in Figure 4B(a)) were subjected to immunostaining using anti-insulin and anti-C-peptide antibodies. In both cases, no massive staining was observed in the tumor mass: only a small cell aggregates appeared to be stained with both antibodies (Appendix A. Notably, cells present at peripheral portion of the tumor were highly reactive to both antibodies, suggesting that they were of mouse (host) origin. These findings suggest the need of in vitro differentiation induction of NSCs toward β-cell lineage for obtaining IPCs efficiently. 

## 4. Discussion

In general, the signaling pathways in pancreatic development during early embryogenesis can be replicated by adding cellular differentiation-related inducers to the medium. Several in vitro protocols that allow iPSCs to differentiate into pancreatic lineage have been reported [18,24,25,26]. However, there are some ESCs and iPSCs that do not always follow the differentiation induction protocols, as described above [17,27]. For instance, when human ESCs were subjected to in vitro differentiation induction toward IPCs using the above-mentioned protocols, the treated human ESCs expressed mature β-cell-specific markers, such as *PDX1*, NK6 homeobox 1 (*NKX6.1*), paired box protein-6 (*PAX-6*), and v-maf musculoaponeurotic fibrosarcoma oncogene family (*MAFA*), together with another endocrine hormone, glucagon, on 18–21 days after differentiation induction. However, these differentiated cells did not correspond to functional pancreatic β-cells because they failed to respond to elevated levels of glucose [17]. Similar observation was also made in this study, wherein the EpiSCs were subjected to in vitro differentiation induction and failed to generate putative iTS-P (see Figure 2B(e,f)). This appeared to be solely due to the limited differentiation potential of EpiSCs. Notably, some reports suggest that EpiSCs inherit epigenetic memory from parental cells [28,29,30,31,32]. Conversion from EpiSCs to NSCs, also known as “naïve conversion” [33], may erase this epigenetic memory. In fact, there are reports that have suggested that during naïve conversion, DNA methylation status in EpiSCs is greatly improved [15]. 

In this study, we examined whether NSCs can be efficiently induced to form pancreatic β-cells (Figure 7). We showed that drug-induced NSCs have the potential to differentiate into pancreatic β-cells when exposed to an appropriate differentiation-inducing stimulation. Similar attempts were also reported by other groups. For examples, Ware et al. demonstrated that NSCs (naïve human ESCs) inclined to differentiate into endodermal lineage, as demonstrated by the expression of liver-specific lineage markers (i.e., albumin, α-fetoprotein, and E-cadherin) and pancreas-specific lineage markers (*PDX1*, *SOX9*, and E-cadherin), when they were forced to form solid tumors (teratomas) in vivo [34]. They speculated that naïve conversion may enhance the ability of cells to differentiate into endoderm in vivo, since an enhanced immunoreactivity for E-cadherin and *PDX1* was observed in the above-mentioned solid teratomas [34]. Furthermore, the increased immunoreactivity was dropped when naïve to primed human ESCs conversion occurred. Park et al. demonstrated that NSCs (naïve human iPSCs) were superior to EpiSCs (primed human iPSCs) with respect to the generation of EBs (10 days after seeding), and were capable of producing CD31 + CD146-expressing vascular progenitors more efficiently [35]. Perhaps, NSCs were more easily induced to produce progenitor or intermediate cells than EpiSCs. 

As shown by Ware et al. [34], in vivo teratoma assay is a powerful tool to evaluate the potential of tumor cells to differentiate and determine their tumorigenic potential. The pre-existing assay largely relies on the subcutaneous grafting or grafting of cells beneath the real capsule [36,37,38]. These approaches often make it impossible for the tumorigenic cells to grow as solid tumors due to the unintentional escape of the grafted cells from the inoculation site. To avoid this, researchers must inoculate a large number of cells. IPPCT is a novel in vivo method in which the cells are injected directly into the pancreatic parenchyma of the immunocompromised mice [19]. As the inoculated cells seldom escape outside, even a small number of tumorigenic cells are able to survive and finally form solid tumors. Furthermore, the differentiation ability of iPSCs injected under the pancreatic environment is not affected. Using IPPCT, we examined the potential of NSCs that had been induced to differentiate into pancreatic β-cell lineage in vitro. As control, NSCs were concomitantly subjected to IPPCT. Grafting of NSCs resulted in the formation of solid tumors (teratomas), measuring <20 mm in diameter, 1.5 months after grafting. On the other hand, there were only small lumps generated 6 months after grafting of NSCs-derived intermediate cells at Stages 3 and 5 (see Figure 4B(a–g) vs. Figure 4B(f–j)). These results suggested loss of tumorigenicity in the cells in the latter groups. These results are in agreement with our previous results in which intermediate cells (iTS-P), generated after partial reprogramming of normal pancreatic mouse cells with Yamanaka factors, never developed into tumors after subcutaneous grafting in immunocompromised mice [17]. As previously noted, in vivo teratoma assay allows the formation of 3-D cell mass, indicated by teratoma formation [19]. This implied that the non-tumorigenic intermediate cells, grafted via IPPCT, form cell mass with highly organized structure. As expected, histological analysis of the whitish small lumps demonstrated the presence of cell mass in both types of explants; however, the lump from the NSCs-derived iTS-P at Stage 5 exhibited more highly organized architecture than that from the NSCs-derived iTS-P at Stage 3 (see Figure 4B(f,g) vs. Figure 4B(h–j)). Immunocytochemical analysis demonstrated that both the cell masses were reactive to anti-insulin (see Figure 5B(a,b,e,f)), and were stained by human cells-specific monoclonal antibody STEM121 (see Figure 5B(c,d,g,h)), suggesting that these cells comprised of cell mass derived from the grafted human cells and were able to produce protein recognized by anti-insulin. Remarkably, there were islet-like structures observed, which were located at the peripheral portion of the pancreas (see Figure 4B(h,j)) and were stained with anti-insulin (see Figure 5B(i,j)). However, these structures were unstained by STEM121 (see Figure 5B(i,k,l)), suggesting that these islet-like cells were of host (mouse) origin. Probably, they were remnant cells that survived under expansion of grafted cells. These results indicated that it is possible to induce highly organized 3-D structures comprising the NSCs-derived intermediate cells in vivo after IPPCT. Further studies are needed to evaluate whether the resulting cell mass can secrete active insulin, leading to cure for diabetic state in Type 1 diabetic model animals, and determine whether it has the ability to respond to an elevated serum glucose level.

## 5. Conclusions

When NSCs were allowed to form EBs, and subsequently subjected to induced differentiation into pancreatic β-cells in vitro, they efficiently generated intermediate cells (assigned to differentiation into β-cells), capable of expressing *PDX1* and insulin. Furthermore, inoculation of these intermediate cells into pancreatic parenchyma of a nude mouse resulted in highly organized 3-D structure, capable of producing proteins recognized by anti-insulin as well as anti-C-peptide antibodies. These findings suggest the possible production of insulin from these 3-D structures. This in vivo grafting also revealed that these intermediate cells are non-tumorigenic. Based on these results, we concluded that NSCs obtained after drug induction of EpiSCs can be used as a source for generating IPCs, and in vivo teratoma assay through IPPCT is a useful system to evaluate the differentiation and tumorigenic potentials of grafted cells. Our future study is now focused on testing whether insulin produced in vivo from the grafted cells has the ability to cure diabetic phenotype of drug-induced nude mice models.

## Figures and Tables

**Figure 1 jcm-09-02838-f001:**
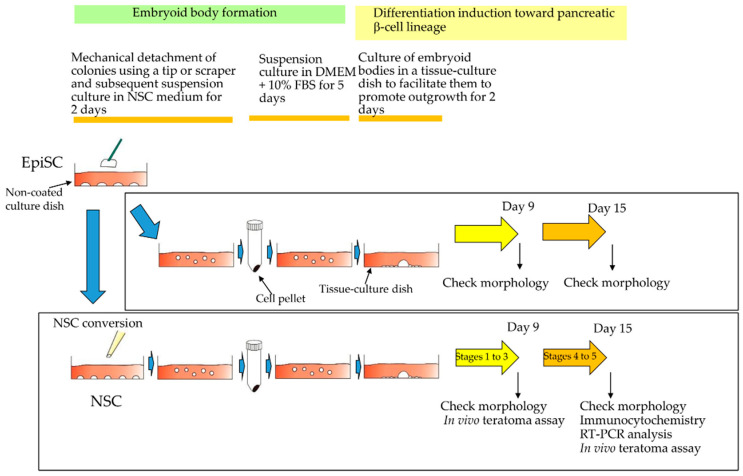
Schematic outline of experimental procedures to examine whether naïve stem cells (NSCs) and/or epiblast stem cells (EpiSCs) have the ability to differentiate into pancreatic β-cells. NSCs and EpiSCs were allowed to form embryoid bodies before being induced to β-cell differentiation in vitro.

**Figure 2 jcm-09-02838-f002:**
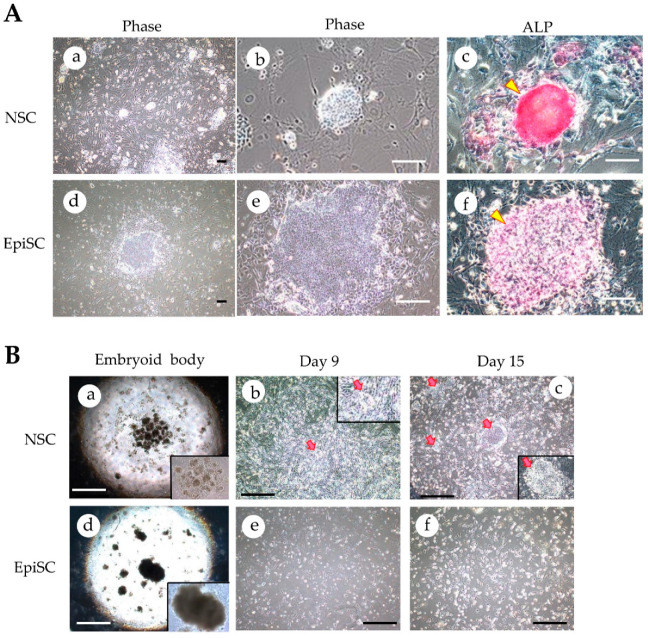
Microscopic observation of naïve stem cells (NSCs), epiblast stem cells (EpiSCs), and their differentiation derivatives. (**A**): Morphology of (a, b, d, e) and alkaline phosphatase (ALP) activity in (c,f) NSCs and EpiSCs. Photographs were taken 5 days after seeding the cells onto mouse embryonic fibroblastic (MEF) cells as feeder cells. ALP activity was measured using a cytochemical staining kit for ALP activity after fixation of cells. Arrowheads indicate the ALP active cells. Bar = 10 μm. (**B**): Morphology of NSCs-derived cells (a–c) or EpiSCs-derived cells (d–f). After embryoid body formation (a,d), the cells were subjected to induction of differentiation into pancreatic β-cells. NSC-derived cells at 9 (b) or 15 (c) days after induction are shown. EpiSCs-derived cells at 9 (e) or 15 (f) days after induction are shown. Arrows indicate putative cobblestone-like morphology of induced tissue-specific stem cells for pancreatic cells (iTS-P). In the right bottom of (a,d), enlarged figures of embryoid bodies are shown. In the right bottom of (c), enlarged figures of iTS-P is shown. Bar = 10 μm.

**Figure 3 jcm-09-02838-f003:**
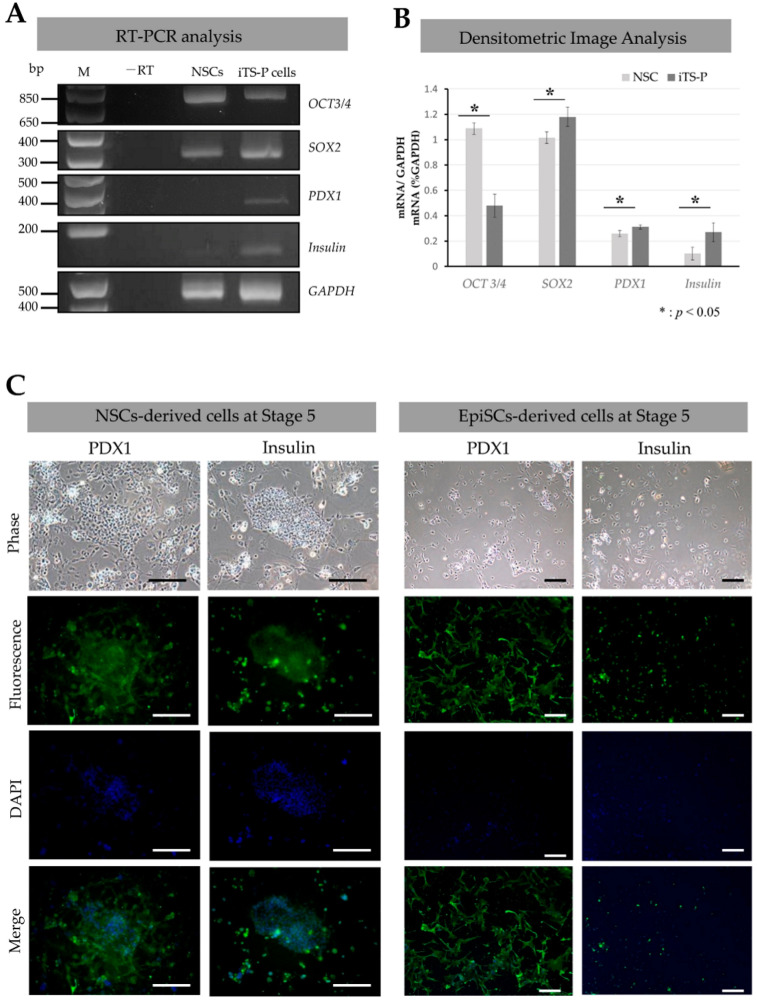
Reverse transcription-polymerase chain reaction (RT-PCR) and immunocytochemical analyses. (**A**): Agarose gel electrophoresis of RT-PCR products derived from naïve stem cells (NSCs) and induced tissue-specific stem cells for pancreatic cells (iTS-P) (collected at 16 days after induced differentiation into pancreatic β-cells). M, 100-bp ladder markers; -RT, no template (water alone) as negative control. (**B**): Comparison of expression levels of each transcript between NSCs and iTS-P cells after normalization. Each band shown in A was scanned using *ImageJ* software. The level of each transcript was normalized with that of *GAPDH* mRNA, and the results obtained are shown in the form of a graph. (**C**): Immunocytochemical staining of NSC-derived cells at Stage 5 and EpiSC-derived cells at Stage 5 with anti-*PDX1* (Green) and anti-insulin (Green) antibodies. Bar = 100 μm.

**Figure 4 jcm-09-02838-f004:**
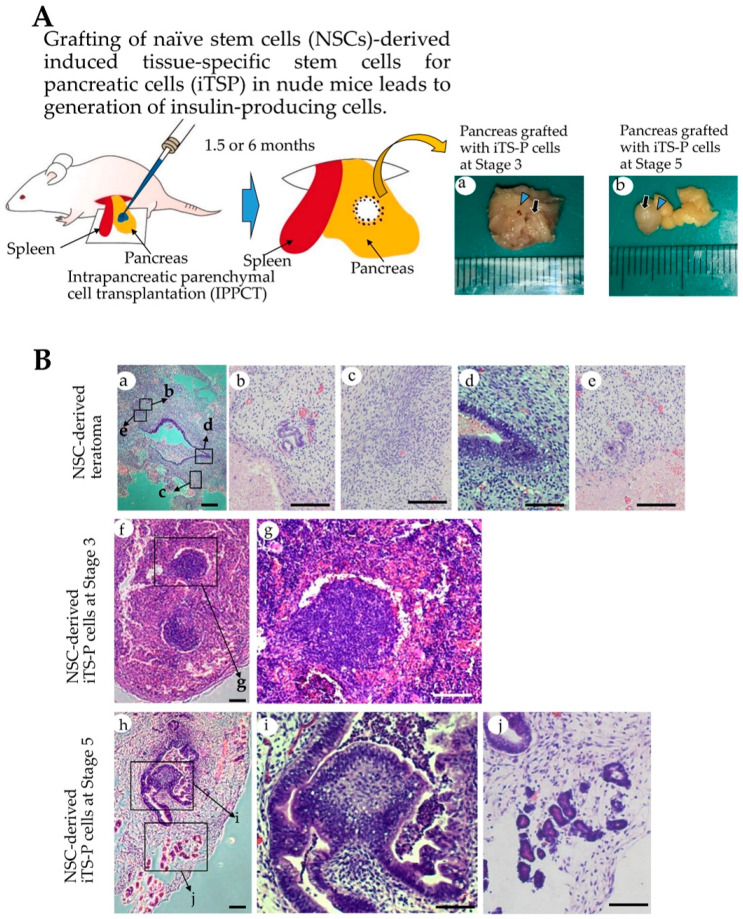
In vivo teratoma formation assay. (**A**): Schematic representation of intrapancreatic parenchymal cell transplantation (IPPCT). Under anesthesia, the spleen and pancreas were pulled out from the left dorsal side of a nude mouse. Cells or cell aggregates were then introduced into pancreatic parenchyma, using a glass micropipette (connected to the mouthpiece) under a dissecting microscope. After growth for 1.5 or 6 months, the growing solid tumor or small lump, generated in the pancreas, were isolated and then subjected to cryostat sectioning. In the right panel, photographs of the pancreas isolated after grafting with induced tissue-specific stem cells for pancreatic cells (iTS-P) at Stage 3 and 5 are shown. Small lump (indicated by arrow), surround by normal-looking pancreas (arrowhead), was discernible in the sample grafted with iTS-P at Stage 3. In the sample grafted with iTS-P at Stage 5, the lump (indicated by arrow) was separated from normal-looking pancreas (arrowhead). (**B**): Histological analysis of the dissected solid tumor or the small lump isolated from the pancreas by hematoxylin-eosin staining (a–e). Solid tumor (teratoma) isolated 1.5 months after inoculation of naïve stem cells (NSCs). Magnified images in b-e are derived from boxes in (a,f,g). Lump isolated 6 months after inoculation of iTS-P at Stage 3. Magnified image in g is derived from a box in (f,h–j). Lump isolated 6 months after inoculation of iTS-P at Stage 5. Magnified images in (i) and (j) are derived from boxes in h. Bar = 100 μm.

**Figure 5 jcm-09-02838-f005:**
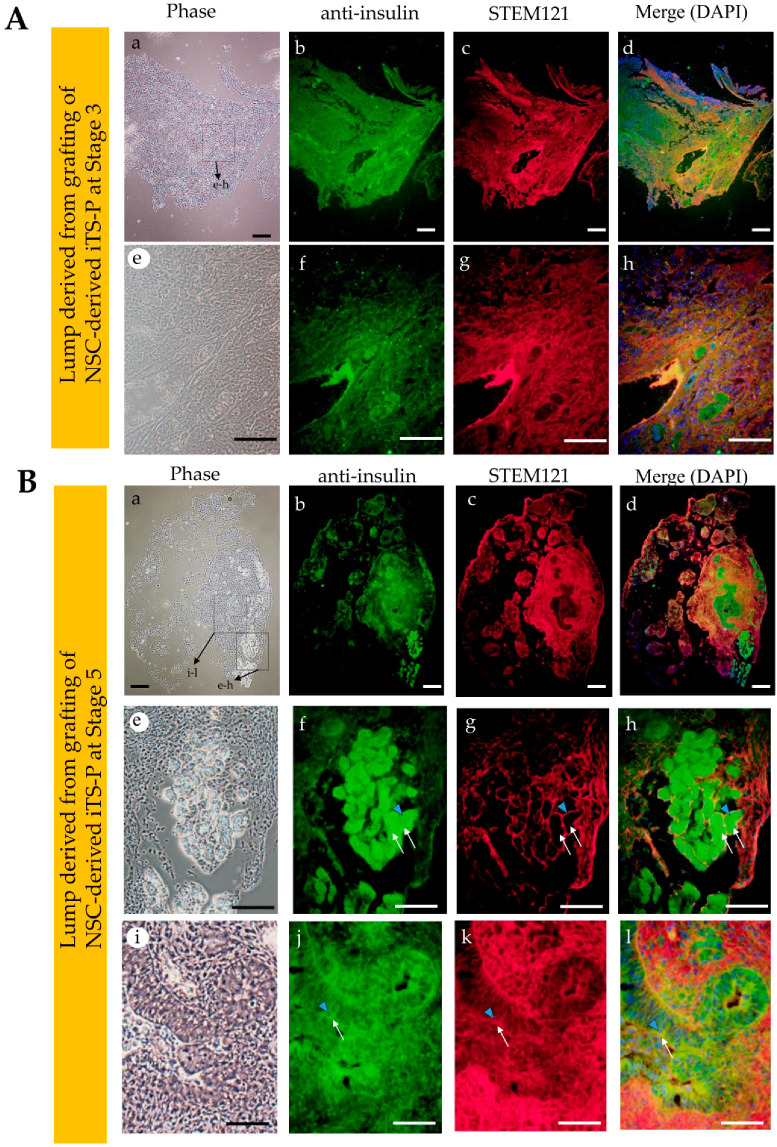
Immunohistochemical staining of cryostat sections stained with anti-insulin (Green) and STEM121 (Red). (**A**): The lump (derived from grafting of induced tissue-specific stem cells for pancreatic cells [iTS-P] at Stage 3) grown in the pancreas of nude mouse (which was located near the side shown in Figure 4B(f,g)), after staining with anti-insulin antibody, and subsequently with human cells-specific monoclonal antibody STEM121. Magnified images in (e–h) are derived from a box shown in (a). Bar = 100 µm. (**B**): The lump (derived from grafting of iTS-P at Stage 5) grown in the pancreas of nude mouse (which was located near the side shown in Figure 4B(h–j)) after staining with anti-insulin antibody, and subsequently with STEM121. Magnified images in (e–h) and (i–l) are derived from boxes shown in (a). The portion shown in a-l was highly reactive to anti-insulin antibody (b). When the glandular cells, which showed extensive staining by anti-insulin antibody (arrowhead in (f,j)), were assessed carefully, it was observed that the intracellular portion of these cells was also stained with STEM121 (arrows in (g,k)), suggesting that these insulin-positive cells are indeed derived from human grafts. In contrast, in the islet-like structure, which was located near the glandular cells, the outer surface of the islets was stained with STEM121 (arrowhead in (g)), but the intracellular portion of these cells was not stained with this antibody (arrows in (g)), suggesting that these islet-structures were of mouse (host) origin. Bar = 100 μm.

**Figure 6 jcm-09-02838-f006:**
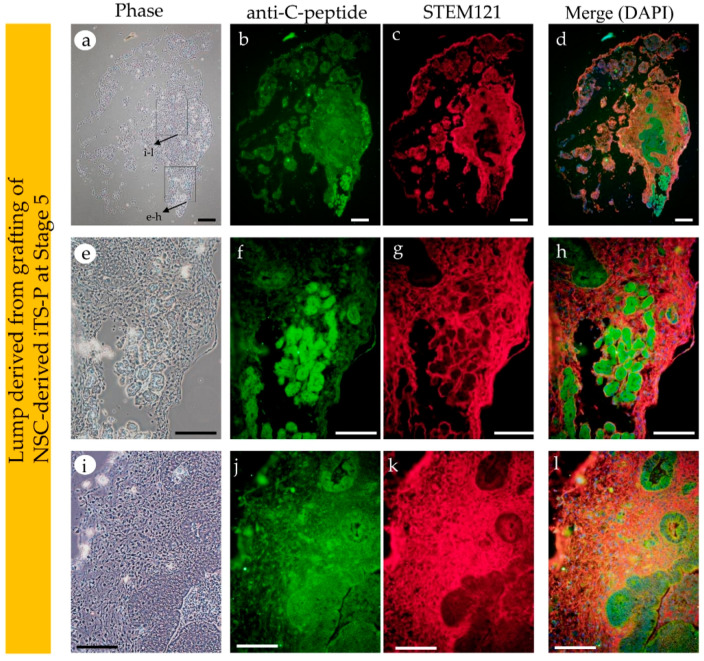
Immunohistochemical staining of cryostat sections stained with anti-C-peptide (Green) and STEM121 (Red). The lump grown in the pancreas of nude mouse after staining with anti-C-peptide antibody, and subsequently with human cells-specific monoclonal antibody STEM121. Magnified images in (**e**–**h**) and (**i**–**l**) are derived from boxes shown in (**a**). The portion shown in (**a**–**d**) corresponds to that shown in B(a-d) of Figure 5. Similarly, the portion shown in (**e**–**h**) corresponds to that shown in Figure 5B(e–h), and the portion shown in (**i**–**l**) corresponds to that shown in Figure 5B(i–l). Notably, the portion stained with anti-insulin antibody was also highly stained by anti-C-peptide antibody. Bar = 100 μm.

**Figure 7 jcm-09-02838-f007:**
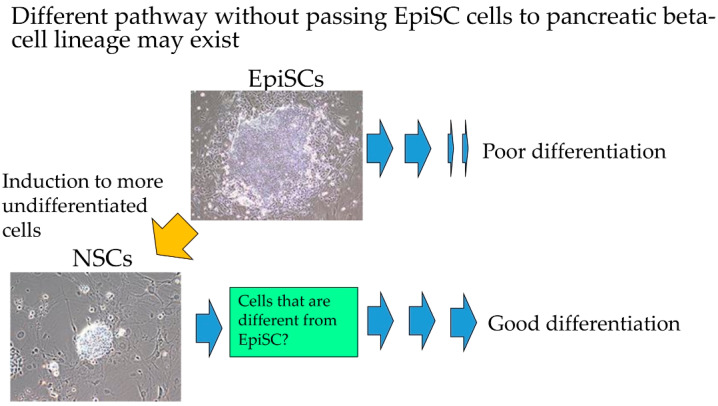
Overall study summary. When epiblast stem cells (EpiSCs) were allowed to differentiate into pancreatic β-cells, the efficiency of generating induced tissue-specific stem cells for pancreatic cells (iTS-P) cells was found to be poor. However, once these cells were converted to the naïve state and then allowed to differentiate, efficient differentiation into iTS-P cells was achieved, with the potential to create β-cells.

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
