# Peer review of "Drug-Induced Naïve iPS Cells Exhibit Better Performance than Primed iPS Cells with Respect to the Ability to Differentiate into Pancreatic β-Cell Lineage"

_jcm, 2020, doi:10.3390/jcm9092838_

Round 1
Reviewer 1 Report
In this paper by Kiyokawa et al., the authors provide an insight into the superior potential of naïve stem cells (NSCs) to differentiate into pancreatic β-cell lineage, compared to primed human stem cells (human induced pluripotent stem cells; iPSCs).
Overall, the data presented here largely supports their hypothesis and conclusions. The work is novel and critical to the field as increasing effectiveness of β-cell differentiation from human stem cells is vital to develop effective therapeutic strategies for diabetes patients in the future. However, it has scope for improvement before being accepted for publication. Suggestions for the authors as follows –
3A: An assumption was made that the Day 15 Epi-SC-derived colonies in Fig. 2B-f may not express pancreatic markers as they did not display the distinct “cobblestone” colony formation seen in Day 15 NSC-derived iTS-P colonies. However, this needs to be conclusively shown and not just by microscopic observation of a phase-contrast image. In order to do this:
- They can include a figure just for Fig2B-f stage cells and run the same conditions shown in Figures 3 and B as part of their Supplementary OR
- Rerun Fig. 3 experiments by including for Fig2B-f stage cells as a condition.
4A: The authors state in the draft that, “As for EpiSCs-derived cells, we failed to generate obvious colonies after in vitro induction of differentiation. This indicated the inability to generate iTS-P when EpiSCs were used as the starting material for isolating pancreatic b-cells. Due to this reason, we did not continue with the grafting of these in vitro cultured EpiSCs derivatives.” In order to confirm this assumption of their inability to generate iTS-P or specifically cells with no pancreatic lineage markers, it is critical to do the experiments described in my previous point above.
4B: Scale bars are not visible in any of the images.
For figures a-e: Are d and e zoomed in images of either a, b, c? Unclear. If so, indicate clearly as has been done in the Figs. 4f-j.
The magnifications are different as well, so scale bars are important, as stated above.
5, 6 and S2: I think the authors have made mistake when they state that 5e-h corresponds to Fig 6e-h and Fig. S2e-h. In my opinion, Fig,5e-h corresponds to Figs, 6i-l and S2i-l instead, based on an overt observation of tissue morphology.
S2: Except a,e & I, label for all other images are missing.
The quality of the images included in the draft is very low with high pixelation. Better quality images should be used in the final draft.
Statistical analysis section in the Methods is missing.
In the abstract, the authors state: “We examined whether NSCs can be efficiently induced to form functional pancreatic β-cells after being subjected to an in vitro protocol.” They are not directly examining functionality here, as glucose response has not been tested. The language should be revised to “potentially functional” or “possibly functional”.
Author Response
Dear Reviewer. Thank you for your comment.
Please see the attachment.

Reviewer 2 Report
The manuscript by Kiyokawa Y et al., examined whether human deciduous teeth dental pulp cells-naïve stem cells (HDDPC-NSCs) can be efficiently induced to form functional pancreatic beta cells in vitro. Overall the manuscript is well written. Based on the data I would remove "functional" as it has not been demonstrated in this paper (abstract and discussion). Demonstrating the effect of beta cell secreted insulin on lowering blood glucose levels in diabetic mice would be considered functional.
Major comments-
- The quality of all the pictures look poor. These should be replaced with better quality images.
- Figure 2A- For a and c provide low magnification images. 2B- It is hard to see cobblestone-like morphology in b. Similar to c authors should include a high magnification inset.
- Figure 3A- Authors should quantify the mRNA levels by RT-qPCR as this will further strengthen their data. 3B- Negative control pictures should be included with the image.
- Page 9, section 3.4- Authors mentioned that NSC-derived iTS-P (~60) or NSCs (~103) were injected to nude mice. How did authors ensured that they are injecting similar number of cells to the nude mice?
- Figures 5 and 6- Authors should include low magnification images. At the moment, just looking at the images it is hard to differentiate between background versus real staining.
- Page 15- Authors mentioned that “using IPPCT methodology, inoculated cells seldom escape outside”. Has this been shown previously e.g. utilizing tracer dye etc?
Author Response
Dear Reviewer 2,
We would like to thank you for your thoughtful consideration of our manuscript entitled, “Drug-induced naïve iPS cells exhibit better performance than primed iPS cells with respect to the ability to differentiate into pancreatic β-cell lineage”. We have revised our manuscript according to your comments. All authors have seen and approved the change.
Major comments-
- The quality of all the pictures look poor. These should be replaced with better quality images.
Answer: Thank you for your comment. We renewed all figures in the revised text.
- Figure 2A- For a and c provide low magnification images. 2B- It is hard to see cobblestone-like morphology in b. Similar to c authors should include a high magnification inset.
Answer: Thank you for your comment. We provided low magnification images at Figure 2A-a and d. We added high magnification insets in Figure 2B-a~d in the revised text.
As suggested by the reviewer, we placed high and low magnification images in Figure 2A in the revised text.
- Figure 3A- Authors should quantify the mRNA levels by RT-qPCR as this will further strengthen their data. 3B- Negative control pictures should be included with the image.
Answer: Thank you for your comment. Based on the data shown in Figure 3A, we quantified the intensity of each band by densitometric scanning and compared expression levels of each gene in a semi-quantitative manner. The results are shown as graphs (please see Figure 3-B in the revised text).
As for the previous Figure 3-B, we checked the presence of PDX1 and insulin in Day 15 EpiSC-derived colonies using an immunocytochemical method and found that these cells expressed pancreatic lineage markers, but were unable to form “cobblestone”-like colony, as shown in Figure 3-C in the revised text. In our previous study, we found that formation of “cobblestone”-like colony is a prerequisite for forming functional cellular mass capable of secreting insulin in vivo. Furthermore, we failed to obtain sufficient numbers of in vitro-differentiated EpiSC-derived cells that are used for cell grafting. These are the reasons why we gave up continuing with the grafting of these in vitro cultured EpiSCs derivatives. These points are mentioned in the revised text (please see L307-316).
- Page 9, section 3.4- Authors mentioned that NSC-derived iTS-P (~60) or NSCs (~103) were injected to nude mice. How did authors ensured that they are injecting similar number of cells to the nude mice?
Answer: Thank you for your comment. In our previous study concerning grafting of iPS cells into the pancreas of a nude mouse, we found that grafting of numerous numbers of NSCs (i.e., >105 cells) resulted in rapid growth of solid tumors (teratomas), leading to occasional death of host individuals within 1.5 months after inoculation. We found that this is ascribed by rapid and extensive propagation of the tumor itself. To avoid occasional death of the host, we had to inoculate a small number of NSCs, from which solid tumors formed within a defined period without causing the host’s death. In contrast, NSC-derived iTS-P grew slowly, which requires inoculation of relatively large number of these differentiated cells for allowing formation of more organized structure.
- Figures 5 and 6- Authors should include low magnification images. At the moment, just looking at the images it is hard to differentiate between background versus real staining.
Answer: Thank you for your comment. We added guide boxes in Figures 5 and 6, which will be helpful for easy identification of target tissues (please see Figure 5 and 6 in the revised text).
- Page 15- Authors mentioned that “using IPPCT methodology, inoculated cells seldom escape outside”. Has this been shown previously e.g. utilizing tracer dye etc?
Answer: Thank you for your comment. We always include trypan blue, a noninvasive vital dye, in the injected solution for monitoring possible leakage of the solution upon IPPCT (Sato et al., IJMS 18: 1678, 2017. DOI: 10.3390/ijms18081678). In most cases, we do not encounter such event. These points are mentioned in the revised text (please see L184-186).

Round 2
Reviewer 1 Report
In the revised draft of their manuscript, Kiyokawa et al have managed to address majority of my comments in a satisfactory manner.
However, here are my follow-up comments which I think will need to be addressed before accepting for publication:
1. My comment for Fig. 4B in the first revision was as follows:
"4B: Scale bars are not visible in any of the images. For figures a-e: Are d and e zoomed in images of either a, b, c? Unclear. If so, indicate clearly as has been done in the Figs. 4f-j. The magnifications are different as well, so scale bars are important, as stated above."
Only the scale bar issue has been addressed. From what I can see, the second part of the comment is not addressed.
2. The authors' comment about not including Statistical Analysis is unsatisfactory. Each paper should statistically analyse their own results and citing a paper which has shown similar results to theirs is not enough. For Fig.3B, they need to conduct a statistical analysis and include a brief section to define how they determined p-values and what p-value was considered significant. Almost every paper has a Statistical Analysis section including the one they cited (10.1371/journal.pone.0163580), so they can them as a guide to writing this section, if necessary.
Author Response
Dear Reviewer 1.
Thank you for your comment.
Please see the attachment.

Reviewer 2 Report
Authors have addressed some of my comments. Listed below is the one that is still missing.
Comment not addressed from previous report-
Remove “functional” as it has not been demonstrated in this paper (abstract and discussion). Demonstrating the effect of beta cells secreted insulin on lowering blood glucose levels in diabetic mice would be considered functional.
New comments-
- Why the DAPI picture for figure 3C (EpiSC-derived cells, insulin) is just showing picture with blue background but no specific cell nuclei?
- What does ~60 and 103 stands for in NSC-derived iTS-P (~60) and NSC (~103)? I was under the assumption that approximately 60 or 1000 cells were injected. But based on author’s reply it seems more cells were injected for NSC-derived iTS-P as these cells were growing slowly compared to NSC.
- Authors should include negative staining controls for figures 5 and 6. At the moment, it is hard to differentiate the real staining from the background. As these are cryostat sections, this could be done very easily.
Author Response
Dear Reviewer 2.
Thank you for your comment.
Please see the attachment.
